# Recent Advances in Imaging Agents Anchored with pH (Low) Insertion Peptides for Cancer Theranostics

**DOI:** 10.3390/molecules28052175

**Published:** 2023-02-26

**Authors:** Yu-Cheng Liu, Zhi-Xian Wang, Jing-Yi Pan, Ling-Qi Wang, Xin-Yi Dai, Ke-Fei Wu, Xue-Wei Ye, Xiao-Ling Xu

**Affiliations:** 1Shulan International Medical College, Zhejiang Shuren University, Hangzhou 310015, China; 2First Clinical College of Traditional Chinese Medicine, Hunan University of Chinese Medicine, Changsha 410208, China

**Keywords:** pH (low) insertion peptide, imaging agents, molecular imaging, cancer theranostics

## Abstract

The acidic extracellular microenvironment has become an effective target for diagnosing and treating tumors. A pH (low) insertion peptide (pHLIP) is a kind of peptide that can spontaneously fold into a transmembrane helix in an acidic microenvironment, and then insert into and cross the cell membrane for material transfer. The characteristics of the acidic tumor microenvironment provide a new method for pH-targeted molecular imaging and tumor-targeted therapy. As research has increased, the role of pHLIP as an imaging agent carrier in the field of tumor theranostics has become increasingly prominent. In this paper, we describe the current applications of pHLIP-anchored imaging agents for tumor diagnosis and treatment in terms of different molecular imaging methods, including magnetic resonance T1 imaging, magnetic resonance T2 imaging, SPECT/PET, fluorescence imaging, and photoacoustic imaging. Additionally, we discuss relevant challenges and future development prospects.

## 1. Introduction

Malignant tumors are among the major diseases that threaten human life, health, and safety. Effectively diagnosing and treating them presents a challenge for the medical community. Most tumors cannot be located based on early symptoms and can only be diagnosed after a certain period, which creates great difficulties for treatment [1]. Therefore, it is important to develop a simple, efficient, and safe method for the early detection of tumors, to enable effective therapeutic intervention.

Molecular imaging is an imaging technology with medical application value that visualizes, characterizes, and measures biological processes at the tissue level, cell level, and subcellular level in vivo using imaging methods. Molecular imaging is critical for the early diagnosis and treatment of diseases, especially tumors. Compared with traditional imaging, molecular imaging can be better used for early tumor screening and diagnosis at the cellular and molecular levels since it exhibits several advantages, including high sensitivity and specificity. Thus, molecular imaging plays an important role in improving the accurate clinical diagnosis and targeted treatment of diseases [2]. Molecular imaging technologies commonly used in the field of tumor imaging include magnetic resonance imaging (MRI), positron emission tomography (PET), single photon emission computed tomography (SPECT), and optical imaging (OI) [3,4]. The application of molecular imaging technology in tumor imaging utilizes the key molecules in the tumor as the recognition target and images the tumor in vivo with the help of specific molecular probes and highly sensitive imaging equipment. At present, gadolinium preparations are the main imaging tracers that are commonly used in clinical practice [5]. However, these contrast agents can easily trigger specific and heterogeneous reactions in the human body and have certain chemical toxicity. At the same time, the agents place the liver and kidney function under pressure. Accordingly, it is an urgent requirement that we develop novel molecular probes consisting of targeting moieties and imaging agents, which can enable imaging agents to actively target tumor tissues, improve diagnostic accuracy, and reduce systemic adverse reactions.

At present, the commonly used molecular probes [6] in clinical practice are antibodies, oligonucleotides [7], and peptides coupled with biotin, fluorescein [8], and radionuclides [9]. Among them, pH (low) insertion peptides (pHLIP) are a kind of peptide that can target the extracellular acid microenvironment or facilitate endosomal release through charge switching and conformational transition [10]. pHLIP-modified drugs or imaging agents can be preferentially delivered to the acid microenvironment at a pH of 6.5, endowing a pH-dependent targeting ability. For ensome escape, pHLIP must be positively charged at the endosomal pH (5.0), which may increase the proton sponge effect and subsequent membrane ruffling. Considerable evidence has indicated that pHLIP may be applied as an imaging carrier and therapeutic agent in the early diagnosis and treatment of tumors [11].

pHLIP is a 36-amino-acid peptide from the third transmembrane helix of bacteriorhodopsin (sequence: NT-GGEQNPIYWARYADWLFTTPLLLLDLALLVDADEGT-C). pHLIP exhibits no definite structure at a neutral pH and mainly uses the random coil conformation [12]. However, at lower pH values, pHLIP changes from a random helix to an α-helical conformation, followed by coupled folding and unidirectional insertion into the membrane bilayer (Figure 1). The pK of membrane insertion of pHLIP is about 6.0. The insertion of pHLIP into the membrane is reversible and unidirectional. In most cases, the N-terminal remains outside the membrane, while the C-terminal translocates through the membrane [13], providing a way for impermeable drugs to transport through the plasma membrane, thus selectively damaging cells with acidic surfaces. The cell surface pH value is related to cell glycolytic activity [14], and glycolytic activity changes with the addition of glucose or deoxyglucose. Due to the high glycolytic activity of metastatic cancer cells, their surface pH is lower than that of nonmetastatic cancer cells; thus, it is possible for pHLIP to insert into the lipid layer when pHLIP is close to the surface. In addition, some researchers have proposed that the affinity of pHLIP to the ether lipid membrane is higher than that to the ester lipid membrane [15]. Compared with ester lipid membranes, the movement restriction and helicity of pHLIP are increased in ether lipid membranes, while ether lipids are upregulated in cancer cells and are biomarkers of various diseases. Therefore, pHLIP should specifically target tumor cells. In addition, tumor vascular endothelial cells are acidic [16]. Research has shown that the membrane insertion principle of pHLIP is mainly related to the molecular structure and dynamic changes of pHLIP near the transition pH value [12]. The corresponding changes include isomerization of the proline-threonine main chain configuration, fracture of the arginine–aspartate bridge, formation of arginine-lipid interactions, and a general decrease in the dynamics of the lipid headgroup and alkyl chain. Some researchers have also suggested that due to the presence of the key residue Asp/Glu, the whole process is triggered by acid-induced protonation [17], so the conformational heterogeneity caused by the strong electrostatic interaction between Asp14 and lipid phosphate groups is the main source of PKA variation [18].

As research into pHLIP has increased, some researchers have found that different ions can enhance or weaken the effect of pHLIP insertion into the cell membrane. Vasquez et al. [19] studied the role of Ca^2+^ and Mg^2+^ in pHLIP insertion, cell targeting, and drug delivery and found that the combination of pHLIP with monomethyl-auristatin-F, a cytotoxic compound, exhibited an increased ability to target HeLa cells (by several times) in the presence of Ca^2+^. Scott et al. [20], meanwhile, found that lipid asymmetry can regulate the formation of an α-helix on the membrane, and Justin et al. [21] found that sodium ions weaken the membrane insertion of the pH (low) insertion peptide.

Many researchers have also tried to modify and adjust the original pHLIP structure, designing a pHLIP variant that exhibits better performance and improved insertion into the lipid membrane. Two positron-emitting radionuclides (^64^Cu and ^18^F) labeled with variants (var3 and var7) and wild-type (WT) pHLIP were used to study the biological distribution of Balb/c mice carrying tumors in situ in 4T1 [22]. As depicted, both ^64^Cu- and ^18^F-labeled var3 showed great tumor accumulation and significant tumor background contrast in images throughout the imaging studies.

Notably, the ability of pHLIP to insert in membranes is regulated by the lipid composition. The central proline substituting for pHLIP exhibits a high helix tendency upon the pH-triggered lipid-dependent conversion of peptide conformation [23].

In summary, as a new type of molecular probe, pHLIP has shown good application prospects in molecular imaging technology for the targeted diagnosis of different types of tumors. This review utilizes different imaging methods as examples to summarize the potential of pHLIP-anchored imaging agents.

## 2. pHLIP-Functionalized Imaging Agents

### 2.1. MRI

Magnetic resonance imaging (MRI) is an imaging technique that uses the signal generated by the resonance of atomic nuclei in the magnetic field to reconstruct the image. Atomic nuclei containing odd-number protons, such as hydrogen nuclei, which are widely found in the human body, are positively charged, and their protons show spin motions and produce magnetic moments, like a small magnet. The spin axes of the small magnets are arranged irregularly. For example, in a uniform strong magnetic field, the spin axes of the small magnets are rearranged according to the direction of the magnetic field line of force. The axes are only arranged in two directions that are parallel or antiparallel to the external magnetic field. The magnets are excited by a specific radio frequency (RF) [24]. As the hydrogen nucleus of the small magnet absorbs a certain amount of energy and resonates, magnetic resonance occurs. When the RF pulse is stopped, the excited hydrogen nucleus gradually releases the absorbed energy, and its phase and energy level return to the state before excitation. This recovery process is called the relaxation process (RP), and the time needed to return to the original equilibrium state is called the relaxation time (RT). There are two kinds of relaxation times. One is the spin-lattice relaxation time [25], or the longitudinal relaxation time, which reflects the time needed for spin nuclei to transfer absorbed energy to the surrounding lattice; this is also the time needed for 90° RF pulse protons to return to the state before longitudinal magnetization excitation after changing from longitudinal magnetization to transverse magnetization, which is known as T1. The other is the spin-spin relaxation time, or the transverse relaxation time, which reflects the process of transverse magnetization attenuation and loss, i.e., the time maintained by transverse magnetization, which is known as T2. T1 and T2 are relatively fixed in normal tissues and pathological tissues in different organs of the human body [26,27], and there are certain differences between them. The difference in relaxation time between tissues is the imaging basis of MRI. Therefore, obtaining T1 or T2 values of various tissues in the selected layer can produce images, including various images of tissue in the layer [28]. In the early stage of clinical MRI applications, many scholars believed that no imaging agent was needed because of the excellent soft tissue resolution. However, with the gradual development of clinical applications, it has been determined that the relaxation times of some tissues overlap, including tumor tissues, making diagnosis by MRI difficult. Furthermore, MRI cannot perform dynamic scanning and determine the functions of organs. Imaging agents can promote a change of energy through dipole-dipole interaction between water molecules and metal ions in the core of contrast agents to a certain extent, thus affecting the recovery of the proton magnetic moment and then indirectly affecting the imaging effect [29]. MRI contrast agents can be divided into positive and negative contrast agents according to the enhancement type [30] as follows: (1) positive contrast agents, which are paramagnetic contrast agents with conventional diagnostic doses, can shorten the T1 relaxation time of the tissue and enhance the signal displayed on the image. Therefore, these contrast agents are called MRI positive contrast agents, an example of which is gadolinium-diethylenetriaminepentaacetic acid (Gd DTPA). (2) Negative contrast agents, meanwhile, exhibit little effect on the T1 relaxation time of the tissue and can shorten the T2 relaxation time, resulting in signal reduction and a dim low signal during TL- or T2-weighted imaging. Superparamagnetic contrast agents belong to this category.

#### 2.1.1. T1 Imaging

The T1 contrast agent is usually a paramagnetic or superparamagnetic metal ion-type contrast agent. Gd DTPA is the earliest and most widely used contrast agent in clinical practice, representing a nonselective extracellular agent. Gd DTPA functions by shortening the T1 relaxation time of tissue [31], increasing the signal intensity of tissue and lesions on T1W1 images to varying degrees, and changing the signal contrast between them to facilitate the detection and diagnosis of lesions. Gd shows a strong paramagnetic effect, but it cannot be injected into the body in free form due to its high toxicity [32]. Gd^3+^ must be prepared in a chelate form or loaded into 13X zeolite nanocarriers to reduce its toxicity [33]. Gd DTPA exhibits high stability and no separation in the organism. Its preparation can be injected into the human body intravenously, but the half-life of excretion through the kidney is only 20 min. With the wide use of Gd DTPA, its shortcomings have become increasingly obvious, such as its low relaxation rate, short cycle time, rapid passage through the cell space after injection, rapid elimination by the kidney, and lack of long-term presence at the lesion site [34]. Gd DTPA is nonbiocompatible, and the toxic effect on cells after dechelation is currently unknown [35]. It has been reported that all Gd-containing contrast agents approved for use in the United States have been associated with cases of nephrogenic systemic fibrosis [36]. Gd can be found deposited in the skin, heart, and kidneys of patients with nephrogenic fibrosis, and even in the human brain, with normal renal function [37,38]. In addition, Gd^3+^ has a particle radius, similar to that of Ca^2+^, which can interfere with calcium-mediated signal pathways; thus, Gd^3+^ is highly toxic [39]. In summary, it is important that we study and develop new, efficient, and safe imaging agents. Janic et al. [40] developed a pH-sensitive paramagnetic nanoparticle (NP) by grafting GdDOTA-4AMP MRI contrast agent and pHLIP onto the surface of the G5 PAMAM dendrimer. pHLIP-coupled Gd44-G5 paramagnetic nanoparticles can bind and fuse with the cell membrane at low pH but do not bind at a normal physiological pH, which can reduce the cytotoxicity of Gd. These nanoparticles can be applied to image different invasive acidic cancer types without other influencing factors. They are reliable imaging contrast agent nanoparticles. Liu et al. [41] prepared pHLIP-anchored GdNPs through a disulfide bond reaction between the pHLIP wild-type sequence containing terminal cysteine residues and AGuIX GdNPs containing sulfhydryl groups (Figure 2), evaluated their radiosensitivity in vitro, and performed magnetic resonance imaging of mice in vivo. In vitro cell uptake experiments showed that the uptake of Gd by pHLIP-anchored GdNPs increased by 78 times. Clone survival experiments showed that the sensitivity of cells to Gd increased by 44% after 5 Gy irradiation at pH 6.2. MRI of living mice showed that compared with that of traditional GdNPs, the retention time of pHLIP GdNPs in tumors was longer (>9 h), the penetrability was better, and the pHLIP GdNPs could penetrate the tumor core area with few blood vessels. This feasibility study confirmed that the combination of pHLIP with GdNPs actively targeted the acidic microenvironment of solid tumors and delivered the original cell-impermeable and radiation-sensitive NPs to cancer cells. This is very important for allowing NP-induced short-range Augerand photons to effectively reach important cell targets, and for supporting the possibility of developing a clinical treatment planning system in the future.

#### 2.1.2. T2 Imaging

Superparamagnetic iron oxide nanoparticles (SPIONs) are a new MRI contrast agent developed after gadolinium. In a key difference from paramagnetic contrast agents, SPIONs can interfere with the uniformity of the local magnetic field because of their high magnetic momentum, so adjacent protons are affected by an “outer region” or “susceptibility effect”, which works to accelerate the desynchronization phase of protons and shorten the T1 and T2 times. However, due to the large particles of these contrast agents, protons are difficult to approach and are affected by unpaired electrons in the inner region, which mainly affect T2 rather than T1 [42]. The influence effect of this kind of contrast agent on T2WI and gradient echo (GE) images is signal reduction, i.e., negative enhancement, so it is called a negative contrast agent. SPIONs exhibit a high magnetic momentum, high magnetic momentum, high relaxation, good biocompatibility, and good blood retention. They are ideal materials to replace gadolinium contrast agents [43]. However, SPIONs cannot be used in inflammatory environments because they are easily excreted. In addition, pure magnetic nanoparticles easily show biofouling effects in blood and thus lose superparamagnetism, resulting in a loss of magnetic resonance signals [44]. SPIO is rich in surface active groups, which easily modify or load targeted molecules, fluorescent dyes, radioisotopes, and drugs. The impact of biological fouling can be removed through surface modifications. Therefore, many studies have been performed on the modification of SPION nanoparticles. SPION nanoclusters modified with pHLIP can show effective pH-responsive retention in different tumor models and tumor-selective imaging in MRI [45], which can provide additional dynamic details of tumor structure and enhance the diagnosis of smaller tumor tissues [46]. Additionally, in contrast to SPIONs, pHLIP-modified SPIONs remained compatible. Magnetite (Fe_3_O_4_) is among the three main iron oxides in the SPION category. Pershina et al. [47] covalently fixed pHLIP to Fe_3_O_4_ magnetic nanoparticles and confirmed that pHLIP-anchored nanocomposites can bind to tumor cells in an acidic environment of HTC cells and LLC tumor models (Figure 3). The biological distribution of nanoparticles in mice and the serum analysis data of experimental animals were further described, demonstrating the clinical potential of pHLIP in targeting the acidic microenvironment of tumors for early detection and MRI diagnosis.

### 2.2. SPECT/PET

SPECT and PET are the most commonly used molecular imaging technologies in nuclear medicine. Both methods have inherent advantages, including their systemic, semiquantitative, and real-time nature, along with their high sensitivity. These methods are among the most important means for accurately diagnosing and treating tumors in clinical practice. SPECT imaging is based on gamma rays, the technique of X-ray development in which antibodies are conjugated to radionuclides (such as ^99m^Tc, ^111^In, ^123^I, and ^177^Lu). With a dedicated gamma-ray detector and a camera or SPECT instrument, the gamma rays can be recorded and converted into an image through signal processing, to determine the location of the radiolabeled antibody. PET imaging requires positron-emitting radionuclides (such as ^18^F, ^68^Ga, ^124^I, and ^89^Zr) for radioactive labeling [48]. The variants formed by the coupling of radionuclides with the noninserted end (N-end) of pHLIP can target the tumor tissue well and also remain in the tumor long enough for the excess and noninserted variants in the normal tissue to be removed, forming high-contrast images [49,50]. Demon et al. [51] used two radionuclides (^64^Cu and ^18^F) to label the pHLIP variants (var3 and var7) derived from NOTA and NO2A and the wild-type (WT) pHLIP to study the biological distribution of 4T1 breast cancer in in situ tumor-bearing mice, and found that all variants formed high-level radioactive markers. Chen et al. [49] synthesized the polypeptide pHLIP (var7)-p1ap using a solid-phase polypeptide synthesis method, labeled pHLIP (var7)-P1AP with ^125^I, observed the biological distribution of ^125^I pHLIP (var7)-P1AP in MDA-MB-231 cells in a breast cancer mouse model and SPECT imaging results in small animals, and found that in an acidic environment, pHLIP (var7)-P1AP could efficiently target MDA-MB-231 cells and inhibit their growth. Small animal SPECT with ^125^I pHLIP (var7)-P1AP could clearly show tumors. A high concentration of ^125^I pHLIP (var7)-P1AP was observed in the tumors, and small animal SPECT imaging showed obvious tumors 4 h after injection. These pHLIP-modified radionuclide variants are currently undergoing clinical transformation research and are expected to enter use as new nuclear imaging tracers for detecting acidic lesions.

### 2.3. Fluorescence Imaging

MRI, SPECT, PET, and other technologies provide effective guidance information for diagnosing cancer, conducting staging, and formulating surgical plans. However, these imaging devices are very large, expensive, complex to operate, and require a large radiation dose, and they mainly provide a focused morphological diagnosis. It remains impossible to accurately diagnose early cancer lesions at the molecular and cellular levels [52]. Fluorescence imaging is a technology that involves falling light illumination. The light source generates a certain wavelength of excitation light from the filter to excite the sample and generate fluorescence; then, a certain wavelength of fluorescence is generated through the filter, returns to the objective lens, and is imaged through the CCD. The technology includes fluorescence reflection imaging (FRI), optical frequency domain imaging (OFDI) [53], fluorescence molecular tomography imaging (FMTI), two-photon fluorescence imaging (TPFI) [54], and so on. As a new noninvasive visualization imaging technology, fluorescence imaging exhibits several advantages, including a fast analysis speed, good biocompatibility, high spatial-temporal resolution, lack of ionizing radiation, and provision of real-time in situ imaging. It is widely used in tumor mechanism research, diagnosis, and treatment, and in surgical navigation [55].

#### 2.3.1. Fluorescence Probes

A fluorescent probe is the core part of fluorescence imaging detection. Its optical properties, such as emission/excitation wavelength, fluorescence quantum yield, sensitivity, and selectivity, determine the efficiency and accuracy of detection. Probes are generally composed of the following important parts: fluorophores and recognition groups [56]. By introducing functional ligands or target groups, these probes can enter specific target organelles, cells, or tissues, and the target-to-background ratio (TBR) of imaging signals is increased. In recent years, fluorescent probes modified by pHLIP have been widely used in targeted research on different tumor model systems. For example, a fluorescent probe formed by the combination of pHLIP with the near-infrared fluorescent dye Alexa750 can target pancreatic ductal adenoma [57]. Golijanina et al. [58] labeled pHLIP with indocyanine green (ICG) and found that ICG pHLIP exhibits a targeting effect on both myometrial invasive and nonmyometrial invasive bladder epithelial cancer. At present, no ICG pHLIP has been shown to have a damaging effect on nontumor tissues. At the same time, studies have shown that ICG pHLIP exhibits an accurate targeting effect on precancerous lesions after radical cystectomy [58]; its sensitivity is up to 97%, and its specificity is up to 100%. Britoa et al. [59] studied a new pH (low) insertion peptide (pHLIP) variant 3 (var3) combined with ICG and found that ICG-var3 pHLIP exhibited high sensitivity and specificity for imaging bladder urothelial carcinoma. This suggests that an ICG imaging agent modified based on pHLIP could be used in the early detection and targeted treatment of bladder cancer. In addition, the biological distribution and tumor targeting of copolymers formed by three kinds of pHLIP variants (variant 3, variant 7, and WT) and fluorescent dyes were detected on 4T1-bearing Balb/c mice [60]. The results showed that the fluorescently labeled pHLIP exhibited an obvious ability to target tumors. The copolymer of Alexa 546 and pHLIP showed the best tumor-targeting effect and lowest accumulation in the liver, kidneys, and muscle, and the fluorescently labeled pHLIP could be removed from tissues with normal pH values after 24 h. Mitrou et al. [61] studied the combination of pH (low) insertion peptide (pHLIP) and fluorescent Alexa532 (Alexa532 pHLIP) and found that it can enhance the fluorescence imaging contrast of pathological breast tissue in vitro. These studies are of great significance for future applications of pHLIP-targeted imaging in clinical research.

#### 2.3.2. Nanoparticle-Based Probe

Although great progress has been made in researching small molecule fluorescent probes in many basic and applied research fields, their residence time at the target site is relatively short, so the imaging time must still be improved. In general, nanoprobes are nanostructured materials modified by specific functional molecules [62]. In a sensing system composed of an identification unit, signal unit, and target, the nanosized probe can realize the direct integration of the signal unit and identification unit, simplify the design and preparation path of the sensing system, and increase the effective area of the identification unit interface. In addition, the probe exhibits several advantages, including high sensitivity, a fast response, and good selectivity, which can overcome the short retention time of ordinary small-molecule fluorescent probes in vivo [63]. pHLIP can mediate the targeted transport of nanoprobes to the surface of tumor cells. For example, pHLIP-coupled upconversion nanoparticles (UCNPs) can successfully target UTUC cells [64]. Ai et al. [65] prepared a pHLIP-coupled UCNP (pHLIP-Ppa-UCNPs), which was formed by a covalent parallel connection of cross-linking agents that can accumulate in a large amount at the tumor site under an acidic condition (pH 6.5). NIR irradiation of breast tumors can effectively kill breast cancer cells, exhibits no obvious toxicity to animals, and has a relatively long half-life. Zhang et al. [66] developed a kind of pHLIP functional nanoprobe with a dual response to drug release, which was used for continuous luminescence imaging and chemotherapy of tumors. They found that the uptake of A549 and HepG2 cells in the acidic extracellular microenvironment (pH < 6.5) was higher than under normal physiological conditions (pH 7.4). The near-infrared continuous luminescence performance of the nanoprobe was clear, and the probe could effectively gather at the tumor site. Thus, visual HepG2 tumor target imaging can be realized without autofluorescence interference. Many other studies have shown that pHLIP-mediated nanoprobes exhibit high specificity and effectiveness and have great potential in tumor detection, diagnosis, and treatment.

### 2.4. Photoacoustic Imaging

PAI is a new imaging method that combines noninvasive optical imaging technology with ultrasonic imaging technology [67]. In PAI, a pulsed laser is used to irradiate biological tissue, and part of the absorbed light energy is converted into heat energy, causing the nearby tissue to undergo thermoelastic expansion; thus, broadband (megahertz level) ultrasonic emission is formed. At this time, the ultrasonic detector located on the surface of the tissue can receive these external ultrasonic waves. According to the detected photoacoustic signal, the image of light energy absorption and distribution in the tissue is reconstructed [68]. PAI breaks through the limitation of low resolution caused by tissue scattering in traditional optical imaging technology and shows high contrast and sensitivity in deep tissue penetration. In view of its unique advantages, PAI has been widely used in tumor system diagnosis and treatment [69]. Since each basic biological component of the image shows different optical absorption spectra, PAI can quickly adjust the excitation to multiple wavelengths so that the distribution of a variety of endogenous or exogenous chromophores can be mapped in vivo. Hemoglobin, myoglobin, melanin, water, lipids, and nucleic acids can all be imaged endogenously in this way. The use of exogenous imaging agents, such as nanoparticles and organic dyes, extends PAI to the field of molecular imaging. Effective exogenous photoacoustic imaging agents can further improve the contrast and resolution of visualization to improve the sensitivity of deep tumor diagnosis [70]. Reshetnyak et al. [71] combined pH-sensitive pHLIP variant 7 with the near-infrared fluorescent dye Alexa750, and based on the analysis of multispectral optoacoustic tomography (MSOT) of human S2VP10 and S2013 pancreatic cancer xenotransplantation model mice, it was shown that the fluorescent probe formed exhibited a targeting effect on pancreatic ductal tumors in the model mice; thus, for the first time, the characterization of the distribution of pH-sensitive targeted probes was achieved in vivo. The study indicates that the combined use of MSOT and pHLIP targeting agents can improve the imaging effect of pancreatic ductal adenocarcinoma. Tian et al. [72] creatively modified gold nanostars (GNs) with pHLIP and successfully designed a pH-responsive multifunctional nanopreparation, GNs pHLIP. MCF-7 cells and their model animals were used to study the in vitro and in vivo targeting of GNs pHLIP. Under pH 6.4 conditions, the cell internalization of GNs pHLIP was twice as high as that of GNs. The accumulation of GNs pHLIP at the tumor site of the model animals was three times that of GNs MPEG. GNS pHLIP also showed a stronger signal than GNs MPEG in PA imaging. It can be seen that GNS pHLIP has good potential application value in tumor imaging and treatment.

## 3. Conclusions

Since the acidic microenvironment is a ubiquitous hallmark of various diseases (tumors, inflammation, ischemic stroke), pHLIP-labeled platforms show good potential for early diagnosis, tumor boundary detection, and evaluation of the therapeutic impact. Recently, a pHLIP-based imaging agent ([^18^F]AlF-cysVar3 pHLIP^®^, NCT04054986) entered a phase 1 clinical trial for patients with breast cancer [73], moving the clinical transformation of pHLIP-based probes forward.

Despite a series of advances, pHLIP-based theranostics still faces many challenges. First, given that hypoxia is primarily involved in causing an acidic tumor microenvironment, an answer to the following must be sought: what is the earliest state of hypoxia that the pHLIP-functionalized imaging agents/probes can detect? Emerging evidence has reported that pHLIPs accumulate in the hypoxic region in correlation with the corresponding microvascular density, that is, when there are more microvessels remaining, there is lower accumulation of pHLIP [74]. However, detailed data about the hypoxic condition are lacking.

Second, the question remains: how can this pHLIP imaging technique distinguish early- from late-stage tumors? Numerous reports have demonstrated that the highly acidic microenvironment is a significant feature in the transition from an avascular preinvasive tumor to a malignant invasive carcinoma [75,76]. Hence, Golijanin et al. [58] compared the identification ability of pHLIP-labeled ICG and ICG-Cys dye. The results revealed that, surpassing ICG-Cys dye alone, the use of pHLIP-labeled ICG identified 22 malignant lesions, including high-grade muscle-invasive urothelial carcinoma in 12, high-grade nonmuscle invasive urothelial carcinoma in 5, carcinoma in situ in 11, and high-grade dysplasia in 1. Nevertheless, the underlying mechanisms must be clarified in future work, and the accuracy distinguished for early- and late-stage tumors.

Third, we must also ask: How effectively are the pHLIP-functionalized probes cleared from the body? Safety considerations necessitate a metabolism study of pHLIP-modified probes. Extant studies pointed out that radiolabeled pHLIP probes can remain intact for at least 1 h in the blood of rats after intraperitoneal injection [77]. The detailed metabolism process in humans must be elucidated systematically.

Once these limitations are overcome, we will be in a position to determine whether pHLIP-based alternatives truly have broad application prospects in the field of molecular imaging.

## Figures and Tables

**Figure 1 molecules-28-02175-f001:**
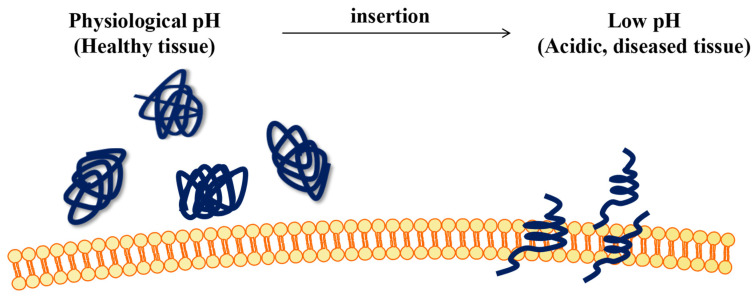
Interaction between pHLIP and cell membrane.

**Figure 2 molecules-28-02175-f002:**
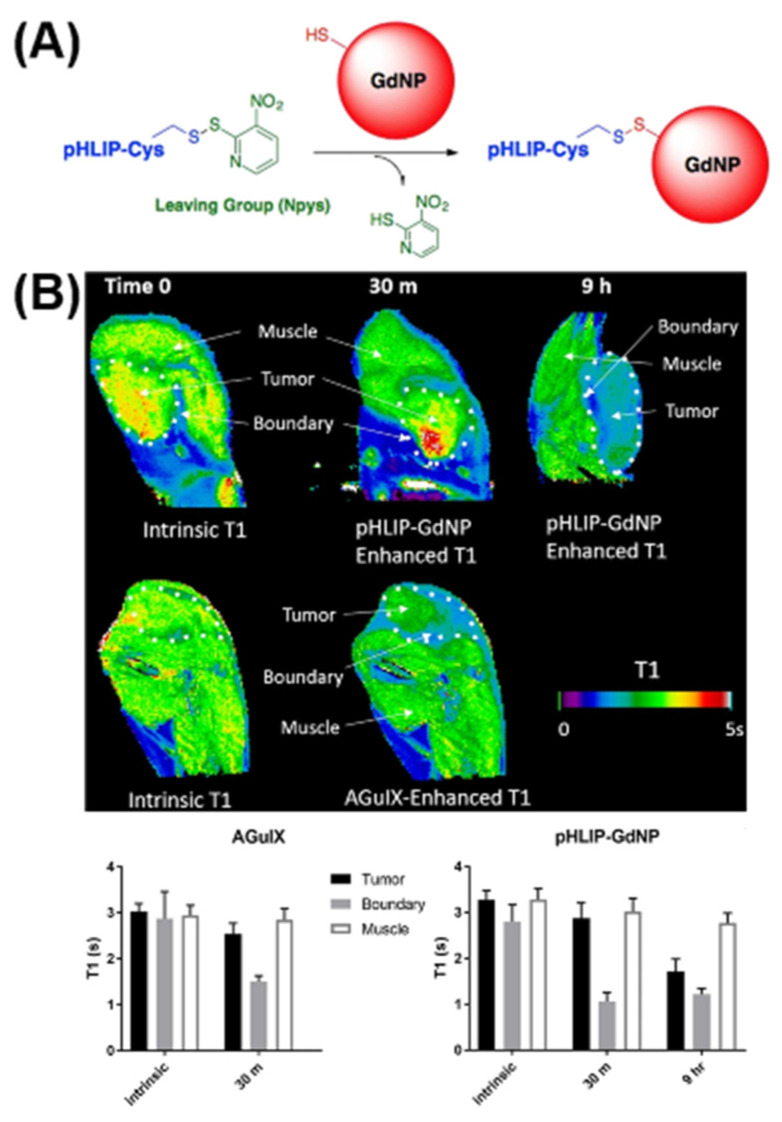
Preparation (**A**) and in vivo imaging (**B**) of pH-anchor gadolinium-based nanoparticles. Reprinted from [41] under the CC BY-NC-ND license.

**Figure 3 molecules-28-02175-f003:**
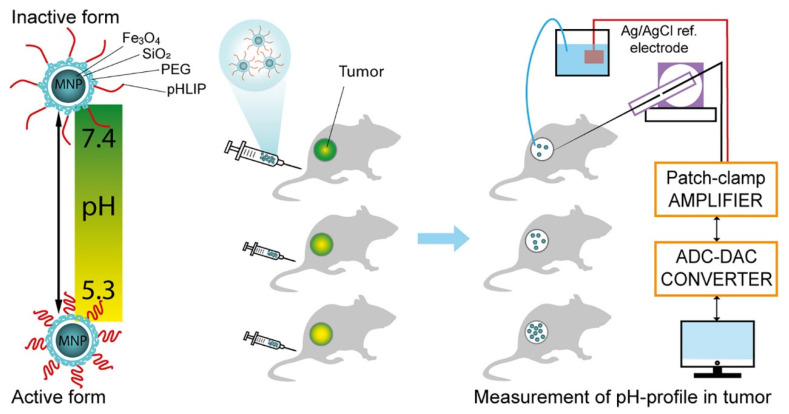
The imaging route of pH-triggered delivery of magnetic nanoparticles conjugated with pH (low) insertion peptide. Reprinted from [47]. Copyright 2022 Elsevier.

## Data Availability

Not applicable.

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
