# Peer review of "Recent Advances in Imaging Agents Anchored with pH (Low) Insertion Peptides for Cancer Theranostics"

_molecules, 2023, doi:10.3390/molecules28052175_

Round 1
Reviewer 1 Report
The current manuscript reviewed the recent advances in imaging agents derived by anchoring pHLIP with improved properties, especially for cancer theragnostic. A pH insertion peptide is capable to form the helix conformation in an acidic microenvironment, thus improves the selectivity and permeability for cancer cells. This manuscript described the applications of pHLIP for tumor diagnosis and potential treatment, especially for different molecular imaging methods, this manuscript should be interesting to the researchers of related studies.
The manuscript was well organized and written, I suggest to accept after very minor modifications:
1. page 3, line 95: "can regulate a the memberane", what does "a" mean?
2. page 7, line 240: "based on r", change to "gamma-ray"
Author Response
Point 1:page 3, line 95: "can regulate a the memberane", what does "a" mean?
Response 1: Thanks so much for your kind suggestion. We felt sorry about our mistake. This sentence had been corrected as below: Scott et al. [20] found that lipid asymmetry can regulate the formation of an α-helix on the membrane. The changes were marked with green in the revision.
Point 2:page 7, line 240: "based on r", change to "gamma-ray"
Response 2: Thank you a lot for your nice comment. Based on your advice, we had changed r" to "gamma-ray". The changes were marked with green. Please check the new submission. Thanks again for your careful review.
Reviewer 2 Report
Manuscript #: Molecules-2176599
Title: Recent Advances in Imaging Agents Anchored with Low pH Insertion Peptides for Cancer Theranostics
Authors: Liu et al
In this review, Liu et al present the current status of imaging agents for cancer theranostics and discuss the utility of a pH-low insertion peptide (pHLIP) and its application in various radio-imaging procedures. The literature has been reasonably well-reviewed, and recent literature has been well-represented. The review covers various types of pHLIP-functionalized imaging agents, their application, challenges, and limitations. To improve the scope of the review, additional information/discussion and some corrections are recommended, as listed below.
1. Introduction: Page 1; Lines 31-32: Instead of “to treat all kinds of tumors,” consider changing to “for effective therapeutic intervention.”
2. Page 2, Line 165: The statement “In addition, the APK app is 6.” This statement appears to be abruptly introduced, and its relevance to the context being discussed is not clear. Revise or removes as appropriate.
3. Page 2, Lines 79-83: Complex sentence. Consider breaking it down to one or more sentences or condensing the content to make it more easily readable.
4. Page 3, Line 90 and Line 98: Instead of stating “some scholars’” consider saying “some studies’…
5. Page 4, Line 165: Clarify which dental acid, as there are several different types. If it is Zeolite/Folic acid, indicate so.
6. Conclusions: Page 9: A brief discussion on the challenges/limitations of the pHLIP agents is needed, especially addressing the following aspects. (a) Given hypoxia is primarily involved in causing the acidic tumor microenvironment, what is the earliest state of hypoxia that the pHLIP-functionalized imaging agents/probes can detect? (b) How can this pHLIP imaging technique distinguish early- and late-stage tumors? (c) How effectively are the pHLIP-functionalized probes cleared from the cells/body, as this is critical for clinical safety?
Author Response
Point 1: Introduction: Page 1; Lines 31-32: Instead of “to treat all kinds of tumors,” consider changing to “for effective therapeutic intervention.”
Response 1: Thank you so much for your kind suggestion. Based on your advice, we had changed “to treat all kinds of tumors” to “for effective therapeutic intervention.”
Point 2: Page 2, Line 165: The statement “In addition, the APK app is 6.” This statement appears to be abruptly introduced, and its relevance to the context being discussed is not clear. Revise or removes as appropriate.
Response 2: Thank you so much for your careful review. We were sorry about our carelessness. We had revised this sentence to “The pK of membrane insertion of pHLIP is about 6.0”.
Point 3: Page 2, Lines 79-83: Complex sentence. Consider breaking it down to one or more sentences or condensing the content to make it more easily readable.
Response 3: Thanks so much for your valuable comment. We had broken the corresponding sentence down to two sentences for improved readability. We sincerely hope the changes will meet your requirements. Thanks again.
Point 4: Page 3, Line 90 and Line 98: Instead of stating “some scholars’” consider saying “some studies’…
Response 4: Thanks so much for your nice comment. We felt sorry about our poor vocabulary and grammar. To make our paper easily readable, we had invited the experts from MDPI to smooth our manuscript. We sincerely hope the revision will be approved by you.
Point 5: Page 4, Line 165: Clarify which dental acid, as there are several different types. If it is Zeolite/Folic acid, indicate so.
Response 5: Thank you for your kind reminder. We had changed our description in the revision as below “Gd3+ must be prepared in a chelate form or loaded into 13X zeolite nanocarriers to reduce its toxicity”.
Point 6: Conclusions: Page 9: A brief discussion on the challenges/limitations of the pHLIP agents is needed, especially addressing the following aspects. (a) Given hypoxia is primarily involved in causing the acidic tumor microenvironment, what is the earliest state of hypoxia that the pHLIP-functionalized imaging agents/probes can detect? (b) How can this pHLIP imaging technique distinguish early- and late-stage tumors? (c) How effectively are the pHLIP-functionalized probes cleared from the cells/body, as this is critical for clinical safety?
Response 5: Thank you so much for this constructive comment, which inspired us deeply. Based on your advice, we had added more discussion in the “Conclusions” part. The changes were marked with yellow highlight. We sincerely hope the new discussion will be appreciated by you. Thanks again for this valuable suggestion.
Reviewer 3 Report
The authors review strategies for cancer research using pHLIP peptide, which respond to pH decrease. As the authors pointed out in the manuscript, such methods may become a powerful tool for cancer targeting. In this manuscript, several applications of the pHLIP peptide are concisely summarized. Thus, this reviewer recommends the paper for publication as it stands.
Minor comment:
Papers using a similar concept with pHLIP are widely reported e.g. by Futaki et al. (Angew. Chem. Int. Ed. 2020, 59, 19990). The authors should discuss such other pH-sensitive peptides in the manuscript.
Typographical error:
1) line65: “APK” should read “PKA”.
Author Response
Papers using a similar concept with pHLIP are widely reported e.g. by Futaki et al. (Angew. Chem. Int. Ed. 2020, 59, 19990). The authors should discuss such other pH-sensitive peptides in the manuscript.
Response: Thank you so much for your kind comment. Based on your suggestion, we add more description about other pH-sensitive peptides in the "Introduction" part of the submitted revision. The corresponding reference was cited. We sincerely hope the changes will be approved by you. Thanks again for your nice advice.